# NEURAL EIGENFUNCTIONS ARE STRUCTURED REPRESENTATION LEARNERS

## ABSTRACT

This paper introduces a structured, adaptive-length deep representation called Neural Eigenmap. Unlike prior spectral methods such as Laplacian Eigenmap that operate in a nonparametric manner, Neural Eigenmap leverages NeuralEF (Deng et al., 2022) to parametrically model eigenfunctions using a neural network. We show that, when the eigenfunction is derived from positive relations in a data augmentation setup, applying NeuralEF results in an objective function that resembles those of popular self-supervised learning methods, with an additional symmetry-breaking property that leads to *structured* representations where features are ordered by importance. We demonstrate using such representations as adaptive-length codes in image retrieval systems. By truncation according to feature importance, our method requires up to $16\times$ shorter representation length than leading self-supervised learning ones to achieve similar retrieval performance. We further apply our method to graph data and report strong results on a node representation learning benchmark with more than one million nodes.

## 1 INTRODUCTION

Automatically learning representations from unlabelled data is a long-standing challenge in machine learning. Often, the motivation is to map data to a vector space where the geometric distance reflects semantic closeness. This enables, for example, retrieving semantically related information via finding nearest neighbors, or discovering concepts with clustering. One can also pass such representations as inputs to supervised learning procedures, which removes the need for feature engineering.

Traditionally, spectral methods that estimate the eigenfunctions of some integral operator (often induced by a data similarity metric) were widely used to learn representations from data (Burges et al., 2010). Examples of such methods include Multidimensional Scaling (Carroll & Arabie, 1998), Laplacian Eigenmaps (Belkin & Niyogi, 2003), and Local Linear Embeddings (Roweis & Saul, 2000). However, these approaches are less commonly employed today than deep representation learning methods that leverage deep generative models or a self-supervised training scheme (Oord et al., 2018; Radford et al., 2018; Caron et al., 2020; Chen et al., 2020a).

There are two primary reasons we believe that contribute to the lesser use of spectral methods today. First, many spectral algorithms operate in a nonparametric manner, such as computing the eigendecomposition of a full similarity matrix between all data points. This makes them difficult to scale to large datasets. Second, the performance of learned representations is highly dependent on the similarity metric used to construct the integral operator. However, picking an appropriate metric for high-dimensional data can itself be a very challenging problem.

In this work, we revisit the approach of using eigenfunctions for representation learning. Unlike past efforts that estimated eigenfunctions in a nonparametric way, we take a different path by leveraging the NeuralEF method (Deng et al., 2022) to parametrically approximate eigenfunctions. Specifically, a deep neural network is trained to approximate dominant eigenfunctions from large-scale data. This learned representation, which we term *Neural Eigenmap*, inherits the principled theoretical motivation of eigenfunction-based representation learning while at the same time gains the flexibility and scalability advantages of deep learning methods.

Our contributions are three-fold:

- We uncover a formal connection between NeuralEF and self-supervised learning (SSL)—applying NeuralEF with a similarity metric derived from data augmentation (Johnson et al., 2022) leads to an objective function that resembles popular self-supervised learning (SSL) methods while also exhibiting an additional symmetry-breaking property. This property enables learning structured representations ordered by feature importance. This ordered structure is lost in other SSL algorithms (HaoChen et al., 2021; Balestriero & LeCun, 2022; Johnson et al., 2022) and gives Neural Eigenmap a key advantage in adaptively setting representation length for best quality-cost tradeoff. In image retrieval tasks, it uses up to 16 times shorter code length than SSL-based representations while achieving similar retrieval precision.

- We show that, even in representation learning benchmarks where the ordering of features is ignored, our method still produces strong empirical performance—it consistently outperforms Barlow Twins (Zbontar et al., 2021), which can be seen as a less-principled approximation to our objective, and is competitive with a range of strong SSL baselines on ImageNet (Deng et al., 2009) for linear probe and transfer learning tasks.

- We establish the conditions when NeuralEF can learn eigenfunctions of indefinite kernels, enabling a novel application of it to graph representation learning problems. On a large-scale node property prediction benchmark (Hu et al., 2020), Neural Eigenmap outperforms classic Laplacian Eigenmap and GCNs (Kipf & Welling, 2016) with decent margins, and its evaluation cost at test time is substantially lower than GCNs.

## 2 NEURAL EIGENFUNCTIONS FOR REPRESENTATION LEARNING

Eigenfunctions are the central object of interest in many scientific and engineering domains, such as solving partial differential equations (PDEs) and the spectral methods in machine learning. Typically, an eigenfunction $\psi$ of the linear operator $T$ satisfies

$$T\psi = \mu\psi, \tag{1}$$

where $\mu$ is a scalar called the eigenvalue associated with $\psi$. In this work, we focus on the kernel integral operator $T_\kappa : L^2(\mathcal{X}, p) \to L^2(\mathcal{X}, p)$,[1] defined as

$$(T_\kappa f)(\boldsymbol{x}) = \int \kappa(\boldsymbol{x}, \boldsymbol{x}') f(\boldsymbol{x}') p(\boldsymbol{x}') \, d\boldsymbol{x}'. \tag{2}$$

Here the kernel $\kappa$ can be viewed as an infinite-dimensional symmetric matrix and thereby Equation (1) for $T_\kappa$ closely resembles a matrix eigenvalue problem.

In machine learning, the study of eigenfunctions and their relationship with representation learning dates back to the work on spectral clustering (Shi & Malik, 2000) and Laplacian Eigenmaps (Belkin & Niyogi, 2003). In these methods, the kernel $\kappa$ is derived from a graph that measures similarity between data points—usually $\kappa$ is a variant of the graph adjacency matrix. Then, for each data point, the outputs of eigenfunctions associated with the $k$ largest eigenvalues are collected as a vector $\psi(\boldsymbol{x}) \triangleq [\psi_1(\boldsymbol{x}), \psi_2(\boldsymbol{x}), \ldots, \psi_k(\boldsymbol{x})]$. These vectors prove to be optimal embeddings that preserve local neighborhoods on data manifolds. Moreover, the feature extractor $\psi_j$ for each dimension is orthogonal to others in function space, so redundancy is desirably minimized. Following Belkin & Niyogi (2003), we call $\psi(\boldsymbol{x})$ the *eigenmap* of $\boldsymbol{x}$.

Our work builds upon the observation that eigenmaps can serve as good representations. But, unlike previous work that solves Equation (1) in a nonparametric way—by decomposing a gram matrix computed on all data points—we approximate $\psi(\boldsymbol{x})$ with a neural network. Our parametric approach makes it possible to learn eigenmaps for a large dataset like ImageNet, meanwhile also enabling straightforward out-of-sample generalization. This is discussed further in the next section.

We leverage the NeuralEF algorithm, proposed by Deng et al. (2022) as a function-space generalization of EigenGame (Gemp et al., 2020), to approximate the $k$ principal eigenfunctions of a kernel using neural networks (NNs). In detail, NeuralEF introduces $k$ NNs $\psi_i, i = 1, ..., k$,[2] which

---

[1] $\mathcal{X}$ denotes the support of observations and $p(\boldsymbol{x})$ is a distribution over $\mathcal{X}$. $L^2(\mathcal{X}, p)$ is the set of all square-integrable functions w.r.t. $p$.

[2] We abuse $\psi_i$ to represent the NN approximating the $i$-th principal eigenfunction if there is no misleading.

are ended with $L^2$-BN layers (Deng et al., 2022), a variant of batch normalization (BN) (Ioffe & Szegedy, 2015), and optimizes them simultaneously by:

$$\max_{\psi_j} R_{j,j} - \alpha \sum_{i=1}^{j-1} R_{i,j}^2 \text{ for } j = 1, \ldots, k \tag{3}$$

where

$$R_{i,j} \triangleq \mathbb{E}_{p(\boldsymbol{x})} \mathbb{E}_{p(\boldsymbol{x}')}[\kappa(\boldsymbol{x}, \boldsymbol{x}')\psi_i(\boldsymbol{x})\psi_j(\boldsymbol{x}')]. \tag{4}$$

In practice, there is no real obstacle for us to use a single shared neural network $\psi : \mathcal{X} \to \mathbb{R}^k$ with $k$ outputs, each approximating a different eigenfunction. In this sense, we rewrite $R$ in a matrix form:

$$R \triangleq \mathbb{E}_{p(\boldsymbol{x})} \mathbb{E}_{p(\boldsymbol{x}')}[\kappa(\boldsymbol{x}, \boldsymbol{x}')\psi(\boldsymbol{x})\psi(\boldsymbol{x}')^\top]. \tag{5}$$

This work adopts this approach as it improves the scaling with $k$ and network size.

Learning eigenfunctions provides a unifying surrogate objective for unsupervised deep representation learning. Moreover, the representation given by $\psi$ is ordered and highly structured—different components are orthogonal in the function space and those associated with large eigenvalues preserve more critical information from the kernel.

## 3 FROM NEURAL EIGENFUNCTIONS TO SELF-SUPERVISED LEARNING

Recent work on the theory of self-supervised learning (SSL) has noticed a strong connection between representations learned by SSL and spectral embeddings of data computed from a predefined augmentation kernel (HaoChen et al., 2021; Balestriero & LeCun, 2022; Johnson et al., 2022). In these works, a clean data point $\bar{\boldsymbol{x}}$ generates random augmentations (views) according to some augmentation distribution $p(\boldsymbol{x}|\bar{\boldsymbol{x}})$. Neural networks are trained to maximize the similarity of representations across different augmentations. Johnson et al. (2022) defined the following augmentation kernel based on the augmentation graph constructed by HaoChen et al. (2021):

$$\kappa(\boldsymbol{x}, \boldsymbol{x}') \triangleq \frac{p(\boldsymbol{x}, \boldsymbol{x}')}{p(\boldsymbol{x})p(\boldsymbol{x}')}, \tag{6}$$

where $p(\boldsymbol{x}, \boldsymbol{x}') \triangleq \mathbb{E}_{p_d(\bar{\boldsymbol{x}})}[p(\boldsymbol{x}|\bar{\boldsymbol{x}})p(\boldsymbol{x}'|\bar{\boldsymbol{x}})]$ and $p_d$ is the distribution of clean data. $p(\boldsymbol{x}, \boldsymbol{x}')$ characterizes the probability of generating $\boldsymbol{x}$ and $\boldsymbol{x}'$ from the same clean data through augmentation, which can be seen as a measure of semantic closeness. $p(\boldsymbol{x}), p(\boldsymbol{x}')$ are the marginal distributions of $p(\boldsymbol{x}, \boldsymbol{x}')$. It is easy to show that this augmentation kernel is positive semidefinite.

Plugging the above definition of $\kappa(\boldsymbol{x}, \boldsymbol{x}')$ into Equation (4) yields

$$R = \mathbb{E}_{p(\boldsymbol{x}, \boldsymbol{x}')}[\psi(\boldsymbol{x})\psi(\boldsymbol{x}')^\top] \approx \frac{1}{B} \sum_{b=1}^{B} \psi(\boldsymbol{x}_b)\psi(\boldsymbol{x}_b^+)^\top. \tag{7}$$

Here, $\boldsymbol{x}_b$ and $\boldsymbol{x}_b^+$ are two independent samples from $p(\boldsymbol{x}|\bar{\boldsymbol{x}}_b)$ with $\bar{\boldsymbol{x}}_1, \bar{\boldsymbol{x}}_2, \ldots, \bar{\boldsymbol{x}}_B$ being a minibatch of data points. Define $\boldsymbol{\psi}_{\mathbf{X}_B} \triangleq [\psi(\boldsymbol{x}_1), \psi(\boldsymbol{x}_2), \cdots, \psi(\boldsymbol{x}_B)] \in \mathbb{R}^{k \times B}$ and $\boldsymbol{\psi}_{\mathbf{X}_B^+}$ similarly. The optimization problems in Equation (3) for learning neural eigenfunctions can then be implemented in auto-differentiation frameworks (Baydin et al., 2018) using a *single* "surrogate" loss—a function that we can differentiate to obtain correct gradients for all maximization problems in Equation (3):

$$\ell(\mathbf{X}_B, \mathbf{X}_B^+) = -\sum_{j=1}^{k} \left(\boldsymbol{\psi}_{\mathbf{X}_B} \boldsymbol{\psi}_{\mathbf{X}_B^+}^\top\right)_{j,j} + \alpha \sum_{j=1}^{k} \sum_{i=1}^{j-1} \left(\hat{\boldsymbol{\psi}}_{\mathbf{X}_B} \boldsymbol{\psi}_{\mathbf{X}_B^+}^\top\right)_{i,j}^2. \tag{8}$$

Here $\hat{\boldsymbol{\psi}}_{\mathbf{X}_B}$ denotes a constant fixed to the value of $\boldsymbol{\psi}_{\mathbf{X}_B}$ during gradient computation, corresponding to the fixed $\psi_i$ involved in the $j$ optimization problem in Equation (3). Throughout this work, we will use the hat symbol to denote a value that is regarded as constant when we are computing gradients. In auto-differentiation libararies, this can be implemented with a stop-gradient operation.

**Learning ordered representations.** As proven by Deng et al. (2022), the above objective function results in each component of $\psi$ converging to a unique eigenfunction ordered by the corresponding eigenvalue. E.g., the first dimension of the output of $\psi$ aligns with the eigenfunction of the largest eigenvalue. This bears similarity to PCA, where the principal components contain most information of the kernel and are orthogonal to each other.

**Linear probe evaluation.** We can view the above optimization problem as a kind of SSL algorithm as it learns representations from mutiple views (augmentations) of data. For SSL methods, a gold standard for quantifying the quality of the learned representations is their linear probe performance, where a linear head is employed to classify the representations to semantics categories. Yet, the linear probe does not take advantage of ordered representations, as suggested by HaoChen et al. (2021) as well. Even if the representation is replaced by the output of an arbitrary span of eigenfunctions, the linear classifier weight can be simply adjusted to produce the same classifier. This implies that replacing $\hat{\psi}_{\mathbf{X}_B}$ with $\psi_{\mathbf{X}_B}$ in Equation (8) (which changes the optimal solution to arbitrary span of eigenfunctions) does not affect the optimal classifier and may actually ease optimization because it relaxes the ordering constraints. So, we adapt the loss specifically for linear probe tasks as follows:

$$\ell_{\text{lp}}(\mathbf{X}_B, \mathbf{X}_B^+) = -\sum_{j=1}^{k} \left(\psi_{\mathbf{X}_B}\psi_{\mathbf{X}_B^+}^{\top}\right)_{j,j} + \alpha \sum_{j=1}^{k}\sum_{i=1}^{j-1} \left(\psi_{\mathbf{X}_B}\psi_{\mathbf{X}_B^+}^{\top}\right)_{i,j}^2. \tag{9}$$

**Connection to Barlow Twins (Zbontar et al., 2021).** Interestingly, the SSL objective defined in Barlow Twins can be written using $\psi_{\mathbf{X}_B}$ and $\psi_{\mathbf{X}_B^+}$:

$$\ell_{\text{BT}}(\mathbf{X}_B, \mathbf{X}_B^+) = \sum_{j=1}^{k} \left[1 - \left(\psi_{\mathbf{X}_B}\psi_{\mathbf{X}_B^+}^{\top}\right)_{j,j}\right]^2 + \lambda \sum_{j=1}^{k}\sum_{i \neq j} \left(\psi_{\mathbf{X}_B}\psi_{\mathbf{X}_B^+}^{\top}\right)_{i,j}^2, \tag{10}$$

where $\lambda$ denotes a trade-off coefficient. This objective makes a close analogy to ours defined in Equation (9). For the first term, our objective directly maximizes diagonal elements, but Barlow Twins pushes these elements to 1. Although they have a similar effect, the gradients and optimal solutions of the two problems can differ. For the second term, we penalize only the lower-diagonal elements while Barlow Twins concerns all off-diagonal ones. With this, we argue the objective of Barlow Twins is an approximation of our objective function for linear probe.

This section builds upon the kernels of HaoChen et al. (2021) and Johnson et al. (2022). The spectral contrastive loss (SCL) of HaoChen et al. (2021) only recovers the subspace spanned by eigenfunctions, so their learned representation does not exhibit an ordered structure as ours. Moreover, as will be shown in Section 6.2, our method empirically benefits more from a large $k$ than SCL. Concurrent to our work, the extended conference version of Johnson et al. (2022) also applied NeuralEF to the kernel of Equation (6) (Johnson et al., 2023). However, they focused on the optimality of the representation obtained by kernel PCA and only tested NeuralEF as an alternative in synthetic tasks. In contrast, our work extends NeuralEF to larger-scale problems such as ImageNet-scale SSL and graph representation learning and discusses the benefit of ordered representation for image retrieval.

## 4 GRAPH REPRESENTATION LEARNING WITH NEURAL EIGENFUNCTIONS

In a variety of real-world scenarios, the observations do not exist in isolation but are related to each other. Their relations are often given as a graph. Assume we have a graph dataset $(\mathbf{X}, \mathbf{A})$, where $\mathbf{X} \triangleq \{x_i\}_{i=1}^{n}$ denotes the node set and $\mathbf{A}$ is the graph adjacency matrix. We define $\mathbf{D} = \text{diag}(\mathbf{A}\mathbf{1}_n)$ and the normalized adjacency matrix $\bar{\mathbf{A}} \triangleq \mathbf{D}^{-1/2}\mathbf{A}\mathbf{D}^{-1/2}$. In spectral clustering (Shi & Malik, 2000), it was shown that the eigenmaps produced by principal eigenvectors of $\bar{\mathbf{A}}$ are relaxed cluster assignments of nodes that minimize the graph cut. This motivates us to use them as node representations in downstream tasks. However, computing these node representations requires eigendecomposition of the $n$-by-$n$ matrix $\bar{\mathbf{A}}$ and hence does not scale well. Moreover, it cannot handle out-of-sample predictions where we need the representation of a novel test example.

We propose to treat $\bar{\mathbf{A}}$ as the gram matrix of the kernel $\dot{\kappa}(x, x')$ on $\mathbf{X}$ and apply NeuralEF to learn its $k$ principal eigenfunctions. However, unlike the augmentation kernel from the last section, the normalized adjacency matrix can be indefinite[3] for an arbitrary graph. Fortunately, we have the following theorem showing the NeuralEF algorithm could still find the $k$ principal eigenfunctions for indefinite kernels as long as it has no less than $k - 1$ positive eigenvalues.

*Theorem* 1 (Extend NeuralEF for processing indefinite kernels). Suppose the kernel $\dot{\kappa}$ has at least $k - 1$ positive eigenvalues. And let

$$R \triangleq \mathbb{E}_{p(x)}\mathbb{E}_{p(x')}[\dot{\kappa}(x, x')\psi(x)\psi(x')^{\top}]. \tag{11}$$

---

[3]One might point out that the graph Laplacian is always positive semidefinite. However, in this case, the eigenmaps should be generated by eigenfunctions with the $k$ *smallest* eigenvalues.

Then, the optimization problem defined in Equation (3) has $\acute{\kappa}$'s $k$ principal eigenfunctions as the solution, of which the $j$-th component is the eigenfunction associated with the $j$-th largest eigenvalue.

We know the normalized adjacency matrix has no less than $k-1$ positive eigenvalues when the graph contains at least $k-1$ disjoint subgraphs (Marsden, 2013), and real-world datasets usually meet this condition. Under this condition, the final surrogate loss for node representation learning using a mini-batch of nodes $\mathbf{X}_B$ as well as the corresponding normalized adjacency $\bar{\mathbf{A}}_B$ is then

$$\ell(\mathbf{X}_B, \bar{\mathbf{A}}_B) = \sum_{j=1}^{k} \left(\boldsymbol{\psi}_{\mathbf{X}_B} \bar{\mathbf{A}}_B \boldsymbol{\psi}_{\mathbf{X}_B}^\top\right)_{j,j} - \alpha \sum_{j=1}^{k} \sum_{i=1}^{j-1} \left(\hat{\boldsymbol{\psi}}_{\mathbf{X}_B} \bar{\mathbf{A}}_B \boldsymbol{\psi}_{\mathbf{X}_B}^\top\right)_{i,j}^2. \tag{12}$$

This makes Neural Eigenmap easily scale up to real-world graphs with millions of nodes.

**Comparison with other graph embedding methods.** Compared to classic nonparametric graph embedding methods like Laplacian Eigenmaps (Belkin & Niyogi, 2003) and node2vec (Grover & Leskovec, 2016), our method enables flexible NN-based out-of-sample prediction. Besides, the training cost of our model is more tolerable than them as they usually entail matrix decomposition whose computational complexity is typically $\mathcal{O}(n^3)$ w.r.t. the number of nodes $n$. Compared to graph neural networks (Kipf & Welling, 2016; Hamilton et al., 2017), our model has substantially faster forward/backward passes, which is especially important for the test phase, because it avoids aggregating information from the graph. Stochastic training is also more straightforward with our method, and the unsupervised nature makes our method benefit from massive unlabeled data.

## 5 RELATED WORK

Self-supervised learning (SSL) has sparked great interest in computer vision. Different methods define different pretext tasks to realize representation learning (Doersch et al., 2015; Wang & Gupta, 2015; Noroozi & Favaro, 2016; Zhang et al., 2016; Pathak et al., 2017; Gidaris et al., 2018). More recent approaches train Siamese nets (Bromley et al., 1993) to model image similarities via contrastive objectives (Hadsell et al., 2006; Wu et al., 2018; Oord et al., 2018; Chen et al., 2020a; He et al., 2020; Caron et al., 2020; Tomasev et al., 2022) or non-contrastive ones (Grill et al., 2020; Chen & He, 2021; Caron et al., 2021; Bardes et al., 2022; Garrido et al., 2022; Bardes et al., 2021). However, due to the existence of trivial constant solutions, popular SSL methods usually introduce empirical tricks such as large batches, asymmetric mechanisms, and momentum encoders to prevent representation collapse. In contrast, Neural Eigenmap removes the requirement for these tricks and builds on more grounded theoretical foundations. We also note that cross-modality representation learning methods like CLIP (Radford et al., 2021) can align the representation space of images and texts and have sparked a variety of practical applications (Shen et al., 2021; Agarwal et al., 2021; Zhou et al., 2021a). Adjusting Neural Eigenmap to cover this kind of contrastive learning deserves further investigation. More recently, transformer-based SSL methods emerge (Bao et al., 2021; Zhou et al., 2021b; He et al., 2022; Assran et al., 2022; Zhou et al., 2022; Fang et al., 2023). They routinely operate on the image patches and usually learn by masked token prediction or its variant.

Theoretical understanding of SSL has gained increasing attention due to the importance of such a learning paradigm. A seminal work by HaoChen et al. (2021) connects contrastively learned representations to the spectral embeddings of the normalized adjacency matrix of an augmentation graph. However, the developed spectral contrastive loss (SCL) only recovers the subspace spanned by eigenfunctions, causing the representation to lose an ordered structure. Subsequently, Johnson et al. (2022) incorporate NT-XEnt and NTLogistic losses into this theoretical framework, but a scalable algorithm for recovering the principal eigenfunctions of the relevant kernel has not been derived. In addition, Balestriero & LeCun (2022) relate two other popular SSL methods, Barlow Twins and VICReg, to spectral analysis methods, and establish a connection between SimCLR and Kernel ISOMAP. Tian (2022) explains contrastive learning as a game between a max player and a min player, and demonstrates a relationship between contrastive losses and PCA for deep linear networks. Furthermore, there have been non-trivial efforts to understand SSL theoretically using techniques beyond spectral learning (Arora et al., 2019; Bansal et al., 2020; Lee et al., 2021; Tian et al., 2020; Tosh et al., 2021; Tsai et al., 2020; Wang & Isola, 2020).

## 6 EXPERIMENTS

In this section, we apply Neural Eigenmap to diverse scenarios to empirically study its behaviors. Neural Eigenmap is easy to implement and we will release the code after acceptance.

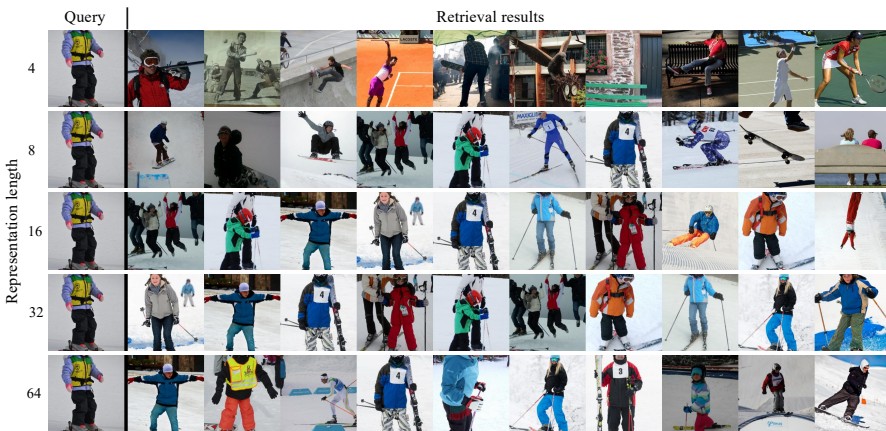

Figure 1: Visualization of retrieval results on COCO with the representations yielded by Neural Eigenmap. Neural Eigenmap is trained on ImageNet and has not been tuned on COCO. The five rows correspond to using the first 4, 8, 16, 32, and 64 entries of the neural eigenmaps for retrieval, respectively. In each row, the first image is a query, and the rest are the top 10 images closest to it over the set.

## 6.1 ADAPTIVE-LENGTH CODES FOR IMAGE RETRIEVAL

Neural Eigenmap learns structured representations where features are ordered by their relative importance. It can be reassuringly truncated without losing critical information of the original data. Here, we exploit this property to perform adaptive compression of representations in image retrieval, where a short code length can significantly reduce retrieval burden (both the memory cost for storage and the time needed to find the top-$M$ closest samples).

We train Neural Eigenmap on ImageNet using the augmentation kernel with the neural eigenfunction defined as a ResNet-50 (He et al., 2016) encoder followed by a 2-layer MLP projector with hidden and output dimension 4096[4] (i.e., $k = 4096$) for **100 epochs**. The augmentation and optimization recipes are identical to those in SimCLR (Chen et al., 2020a). We set $\alpha = 0.005$ for $k = 4096$ and linearly scale it for other values of $k$ (e.g., when $k = 8192$, we set $\alpha$ to 0.0025). After training, we evaluate the learned representations on COCO, NUS-WIDE (Chua et al., 2009), PASCAL VOC 2012 (Everingham et al.), and MIRFLICKR-25000 (Huiskes & Lew, 2008) by performing image retrieval based on standard data splits. We highlight that *no further fine-tuning is performed*.

Images whose representations have the largest cosine similarity with the query ones are returned. We evaluate the results by mean average precision (mAP) and precision with respect to the top-$M$ returned images. We set $M = 5000$ for COCO and NUS-WIDE, and set $M = 100$ for PASCAL VOC 2012 and MIRFLICKR-25000. The returned images are considered to be relevant to the query image when at least one class labels of them match. We include Neural Eigenmap w/o `stop_grad` and SCL as two baselines because (*i*) the comparison between Neural Eigenmap and Neural Eigenmap w/o `stop_grad` can reflect that learning ordered eigenfunctions leads to structured representations; (*ii*) SCL is effective in learning representations as revealed by previous studies and is also related to spectral learning. We experiment with codewords of various lengths. For Neural Eigenmap, we use the elements with small indices in the representations. The representations of Neural Eigenmap w/o `stop_grad` and SCL are non-structured, so we randomly sample elements to perform retrieval and report the average results and error bars over 10 runs.

We present the results in Figure 2 and Figure 6 in Appendix. As shown, Neural Eigenmap requires up to $16\times$ fewer representation dimensions than the competitors to achieve similar retrieval performance. We also note that the retrieval performance of Neural Eigenmap drops when the code length is too high. This is probably because the NeuralEF objective has trouble recovering the eigenfunctions associated with small eigenvalues (Deng et al., 2022), so the tailing components may contain useless information. A potential solution is to add perturbations to the kernel to remove small eigenvalues, making all eigenfunctions more accurately recoverable. We leave this as future work.

We further visualize some retrieval results on COCO in Figure 1. They are consistent with the quantitative results. We can see the results quickly become satisfactory when the code length exceeds

---

[4]We apply batch normalization (Ioffe & Szegedy, 2015) and ReLU to the hidden layer.

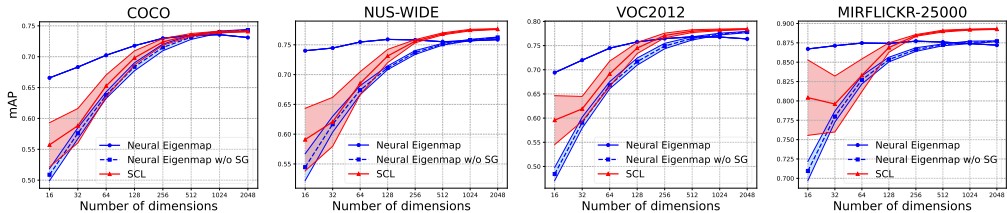

Figure 2: Retrieval mAP varies w.r.t. representation dimensionality.

Table 2: ImageNet linear probe accuracy varies w.r.t. batch size. All methods use a 2-layer MLP projector of dimension 2048. BT refers to Barlow Twins.

| Batch size | *SCL* | *BT* | *Neural Eigenmap* |
|---|---|---|---|
| 256 | **63.0** | 57.6 | 60.5 |
| 512 | **64.6** | 60.3 | 63.3 |
| 1024 | 65.6 | 61.8 | **65.7** |
| 2048 | 66.5 | 60.4 | **66.8** |

Table 3: ImageNet linear probe accuracy varies w.r.t. the dimension of the 2-layer MLP projector. All methods adopt a batch size of 2048.

| Projector dim. | *SCL* | *BT* | *Neural Eigenmap* |
|---|---|---|---|
| 2048 | 66.5 | 60.4 | **66.8** |
| 4096 | 67.1 | 63.9 | **67.7** |
| 8192 | NaN | 66.2 | **68.4** |

8, which implies that the first few elements of our representations already contain rich semantics of the input. Refer to Appendix B.3 for the comparison between our methods and another baseline that combines SCL and principal component analysis (PCA).

## 6.2 UNSUPERVISED VISUAL REPRESENTATION LEARNING

**Linear Probe.** We follow the setups of Section 6.1. We train a supervised linear classifier on the representations yielded by the ResNet-50 encoder and then test it. We compare to popular SSL methods including SimCLR (Chen et al., 2020a), SwAV (Caron et al., 2020), MoCo v2 (Chen et al., 2020b), BYOL (Grill et al., 2020), SimSiam (Chen & He, 2021), spectral contrastive loss (SCL) (HaoChen et al., 2021), and Barlow Twins (Zbontar et al., 2021), with the results reported in Table 1. As shown, Neural Eigenmap can beat all baselines. The performance gain of Neural Eigenmap over Barlow Twins reflects the merits of our formulation. We note that SimSiam, with a batch size of 256, is also well-performing, so it may be preferred when resources are constrained. Yet, the smaller batch size would substantially increase the training time. Our method should be preferred when the memory cost is not a concern, such as on a standard lab server with multiple GPUs.

Table 1: Comparisons on ImageNet linear probe accuracy (%) with the ResNet-50 encoder pre-trained for *100 epochs*. The results of Sim-CLR, SwAV, MoCo v2, BYOL, and SimSiam are from (Chen & He, 2021). The result of SCL is from (HaoChen et al., 2021), and that of Barlow Twins is reproduced by ourselves. As shown, our method outperforms all baselines.

| Method | batch size | top-1 accuracy |
|---|---|---|
| *SimCLR* | 4096 | 66.5 |
| *SwAV* | 4096 | 66.5 |
| *MoCo v2* | 256 | 67.4 |
| *BYOL* | 4096 | 66.5 |
| *SimSiam* | 256 | 68.1 |
| *SCL* | 384 | 67.0 |
| *Barlow Twins* | 2048 | 66.2 |
| *Neural Eigenmap* | 2048 | **68.4** |

SCL and Barlow Twins deploy similar learning objectives with Neural Eigenmap, so we opt to take a closer look at their empirical performance.[5] We reproduce them to place them under the same training protocol as Neural Eigenmap for a fair comparison. In particular, we have tuned the trade-off hyper-parameter $\lambda$ in Barlow Twins, which plays a similar role with the $\alpha$ in our method. We present the results in Table 2 and Table 3. When fixing the hidden and output dimension of the projector as 2048, we see that increasing batch size enhances the performance of all three methods (except for the batch size 2048 for Barlow Twins). Compared to the other two methods, a medium batch size like 1024 or 2048 can yield significant gains for Neural Eigenmap. Meanwhile, when fixing the batch size as 2048, all methods yield better accuracy when using a higher projector dimension (but

---

[5]VICReg borrows the covariance criterion from Barlow Twins and the two methods perform similarly, so we only include Barlow Twins into our studies.

Table 4: Transfer learning on COCO detection and instance segmentation. All unsupervised methods are pre-trained on ImageNet for 200 epochs using ResNet-50. Mask R-CNNs (He et al., 2017) with the C4-backbone (Girshick et al., 2018) are built given the pre-trained models and fine-tuned in COCO 2017 train (1× schedule), then evaluated in COCO 2017 val. The results of the competitors are from Chen & He (2021).

| Pre-training method | COCO detection | | | COCO instance seg. | | |
|---|---|---|---|---|---|---|
| | $AP_{50}$ | AP | $AP_{75}$ | $AP_{50}^{mask}$ | $AP^{mask}$ | $AP_{75}^{mask}$ |
| *ImageNet supervised* | 58.2 | 38.2 | 41.2 | 54.7 | 33.3 | 35.2 |
| *SimCLR* | 57.7 | 37.9 | 40.9 | 54.6 | 33.3 | 35.3 |
| *MoCo v2* | 58.8 | 39.2 | 42.5 | 55.5 | 34.3 | 36.6 |
| *BYOL* | 57.8 | 37.9 | 40.9 | 54.3 | 33.2 | 35.0 |
| *SimSiam, base* | 57.5 | 37.9 | 40.9 | 54.2 | 33.2 | 35.2 |
| *SimSiam, optimal* | 59.3 | 39.2 | 42.1 | 56.0 | 34.4 | 36.7 |
| *Barlow Twins* | 59.0 | 39.2 | 42.5 | 56.0 | 34.3 | 36.5 |
| *Neural Eigenmap* | **59.6** | **39.9** | **43.5** | **56.3** | **34.9** | **37.4** |

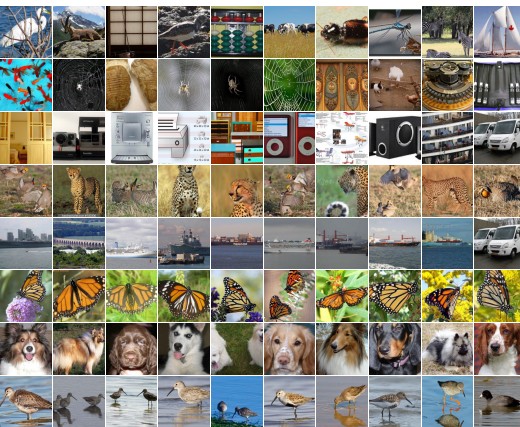

Figure 3: The top 10 samples from the ImageNet validation set that predominantly excite the first 8 principal neural eigenfunctions.

SCL failed to converge when setting the dimension to 8192). We can see that NEigenmap benefits more from a higher output dimension than SCL.

We next study if a longer training procedure would result in higher linear probe performance. We compare Neural Eigenmap to SimSiam because it is strongest baseline in Table 1. The results are shown in Table 5. Neural Eigenmap consistently outperforms SimSiam as we increase the training epochs.

Table 5: Comparisons on ImageNet linear probe accuracy with various training epochs.

| Method | 100 ep | 200 ep | 400 ep |
|---|---|---|---|
| *SimSiam* | 68.1 | 70.0 | 70.8 |
| *Neural Eigenmap* | 68.4 | 70.3 | 71.5 |

**Transfer Learning.** We then evaluate the representation quality by transferring the features to object detection and instance segmentation tasks on COCO (Lin et al., 2014). The models are pre-trained for 200 epochs and then fine-tuned end-to-end on the target tasks following standard practice. We base our experiments on the public codebase from MoCo[6] (He et al., 2020) and tune only the fine-tuning learning rate (and set it to 0.05) as suggested by Chen & He (2021). The results in Table 4 demonstrate that Neural Eigenmap has better transferability than existing approaches. It achieves leading results across tasks and metrics with clear gaps.

## 6.3 VISUALIZATION OF THE LEARNING EIGENFUNCTIONS

We visualize the neural eigenfunctions learned on ImageNet by examining which samples predominantly excite them. In Figure 3, we present the top 10 samples from the validation set that elicit the strongest responses for the first 8 neural eigenfunctions. An interesting observation is that samples

---

[6]https://github.com/facebookresearch/moco.

Table 6: Comparisons on OGBN-Products test accuracy (%). The results of Neural Eigenmap refer to the linear probe performance. The results of the baselines are based on non-linear classifiers.

| Method | 100% training labels | 10% training labels | 1% training labels |
|---|---|---|---|
| *Plain MLP* | $62.16 \pm 0.15$ | $57.44 \pm 0.20$ | $47.76 \pm 0.62$ |
| *Laplacian Eigenmap + MLP* | $64.21 \pm 0.35$ | $58.99 \pm 0.20$ | $49.94 \pm 0.30$ |
| *Node2vec + MLP* | $72.50 \pm 0.46$ | $68.72 \pm 0.43$ | $61.97 \pm 0.44$ |
| *GCN* | $75.72 \pm 0.31$ | $73.14 \pm 0.34$ | $67.61 \pm 0.48$ |
| *Neural Eigenmap* | $\mathbf{78.33} \pm 0.08$ | $\mathbf{75.78} \pm 0.46$ | $\mathbf{68.04} \pm 0.39$ |

within the same row exhibit similar semantic structures, while variations between the rows suggest potential orthogonality among the learned neural eigenfunctions. More results are in Appendix B.4.

## 6.4 NODE REPRESENTATION LEARNING ON GRAPHS

We then apply Neural Eigenmap to OGBN-Products (Hu et al., 2020), one of the most large-scale node property prediction benchmarks, with $2,449,029$ nodes and $61,859,140$ edges. We omit small-scale benchmarks since a large abundance of nodes are particularly important for Neural Eigenmap to learn generalizable representations. We use the graph kernel and specify the neural eigenfunction with a 11-layer MLP encoder followed by a projector. We set the encoder width to 2048 and equip it with residual connections (He et al., 2016) to ease optimization. The projector is identical to that in Section 6.2. The training is performed on all nodes for 20 epochs using a LARS (You et al., 2017) optimizer with batch size $16384$, weight decay 0, and learning rate 0.3 (accompanied by a cosine decay schedule). We tune $\alpha$ according to the linear probe accuracy on validation data and finally set it to 0.3.

After training, we assess the representations yielded by the encoder with linear probe. The training of the linear classifier lasts for 100 epochs under a SGD optimizer with batch size 256, weight decay $10^{-3}$, and learning rate $10^{-2}$ (with cosine decay). We experiment with varying numbers of training labels for performing linear probe to examine representation quality systematically. We compare to two non-parametric node embedding approaches, Laplacian Eigenmap and node2vec: the computed node embeddings are augmented to node features, on which MLP classifiers are trained. We include two other baselines GCN and MLP, which are directly trained on raw node features. We base the implementation on the public codebase[7]. MLP baselines all have three layers of width $512$, and it is empirically observed larger width cannot bring considerable gains.

Table 6 displays the comparison on test accuracy (summarized over 10 runs). Neural Eigenmap has shown superior performance over the baselines across multiple settings. *Laplacian Eigenmap + MLP* underperforms Neural Eigenmap because the representations yielded by Laplacian Eigenmap contain only undecorated spectral information of the graph Laplacian, while the representations of Neural Eigenmap are *the outputs of the encoder*, which correspond to a kind of harmonized Laplacian Eigenmap according to the node features. Nevertheless, one limitation of Neural Eigenmap is that its training cost is substantially higher than the baselines (due to the large encoder).

**Test cost.** In the test phase, Neural Eigenmap makes predictions through a forward pass, while GCN still needs to aggregate information from the graph. Therefore, Neural Eigenmap is more efficient than GCN at test time— GCN's prediction time for a test datum is 0.3818s, while for Neural Eigenmap this is 0.0013s (on an RTX 3090 GPU).

## 7 CONCLUSION

In this paper, we formulate unsupervised representation learning as training neural networks to approximate the principal eigenfunctions of a pre-defined kernel. Our learned representations is structured—features with smaller indices contain more critical information. This is a key advantage that distinguishes our work from existing self-supervised learning methods. We provide strong empirical evidence of the effectiveness of our structured representations on large-scale benchmarks. Future directions may include designing suitable kernels for other data modalities such as video, image-text pairs, and point clouds.

---

[7] https://github.com/snap-stanford/ogb/tree/master/examples/nodeproppred/products.

## REPRODCIBILITY STATEMENTS

We submit the code for reproducing the results of image retrieval and linear probe. Please refer to README.md for specific instructions.

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

# A  PROOF

## A.1  PROOF OF THEOREM 1

*Lemma* 1. Let $\mu_j$ denote the eigenvalues of $\dot\kappa$ and $\delta$ the indicator function. Let $\mu_s \triangleq \inf_{j\geq 1}\mu_j$ and assume $\mu_s > -\infty$. The kernel $\kappa(\boldsymbol{x}, \boldsymbol{x}') \triangleq \dot\kappa(\boldsymbol{x}, \boldsymbol{x}') - \mu_s \delta_{\boldsymbol{x}=\boldsymbol{x}'}/\sqrt{p(\boldsymbol{x})p(\boldsymbol{x}')}$ is positive semidefinite and has the same eigenfunctions as $\dot\kappa(\boldsymbol{x}, \boldsymbol{x}')$.

*Proof.* Let $(\nu_j, \psi_j)$ denote an eigenpair of $\kappa(\boldsymbol{x}, \boldsymbol{x}')$. By the definition of eigenfunction, we have

$$\int \kappa(\boldsymbol{x}, \boldsymbol{x}')\psi_j(\boldsymbol{x}')p(\boldsymbol{x}')d\boldsymbol{x}' = \nu_j\psi_j(\boldsymbol{x}).$$

It follows that

$$\int \dot\kappa(\boldsymbol{x}, \boldsymbol{x}')\psi_j(\boldsymbol{x}')p(\boldsymbol{x}')d\boldsymbol{x}' = \int \kappa(\boldsymbol{x}, \boldsymbol{x}')\psi_j(\boldsymbol{x}')p(\boldsymbol{x}')d\boldsymbol{x}' + \mu_s \int \frac{\delta_{\boldsymbol{x}=\boldsymbol{x}'}}{\sqrt{p(\boldsymbol{x})p(\boldsymbol{x}')}}\psi_j(\boldsymbol{x}')p(\boldsymbol{x}')d\boldsymbol{x}'$$

$$= \nu_j\psi_j(\boldsymbol{x}) + \frac{\mu_s}{\sqrt{p(\boldsymbol{x})}}\int \delta_{\boldsymbol{x}=\boldsymbol{x}'}\sqrt{p(\boldsymbol{x}')}\psi_j(\boldsymbol{x}')d\boldsymbol{x}'$$

$$= \nu_j\psi_j(\boldsymbol{x}) + \frac{\mu_s}{\sqrt{p(\boldsymbol{x})}}\sqrt{p(\boldsymbol{x})}\psi_j(\boldsymbol{x})$$

$$= (\nu_j + \mu_s)\psi_j(\boldsymbol{x}).$$

Namely, $(\nu_j + \mu_s, \psi_j)$ is an eigenpair of $\dot\kappa(\boldsymbol{x}, \boldsymbol{x}')$. Since $\mu_s$ is the smallest eigenvalues of $\dot\kappa(\boldsymbol{x}, \boldsymbol{x}')$, we have $\nu_j + \mu_s \geq \mu_s$, then $\nu_j \geq 0$. Therefore, any eigenvalue of $\kappa(\boldsymbol{x}, \boldsymbol{x}')$ is non-negative.

Similar to the above, it is easy to show that any eigenfunction of $\dot\kappa(\boldsymbol{x}, \boldsymbol{x}')$ will also be the eigenfunction of $\kappa(\boldsymbol{x}, \boldsymbol{x}')$, with eigenvalues shifted by $-\mu_s$. Therefore, we conclude that the two kernels have the same eigenfunctions. □

*Theorem* 1. Suppose the kernel $\dot\kappa$ has at least $k - 1$ positive eigenvalues. And let

$$R \triangleq \mathbb{E}_{p(\boldsymbol{x})}\mathbb{E}_{p(\boldsymbol{x}')}[\dot\kappa(\boldsymbol{x}, \boldsymbol{x}')\psi(\boldsymbol{x})\psi(\boldsymbol{x}')^\top]. \tag{11}$$

Then, the optimization problem defined in Equation (3) has $\dot\kappa$'s $k$ principal eigenfunctions as the solution, of which the $j$-th component is the eigenfunction associated with the $j$-th largest eigenvalue.

*Proof.* We reuse the notations in Lemma 1. When $\mu_s \geq 0$, the kernel is positive semidefinite and the result follows directly from Deng et al. (2022, Theorem 1 and Eq. (14)). We prove the $\mu_s < 0$ case in the following.

Denote by $\psi_j : \mathcal{X} \rightarrow \mathbb{R}$ the function corresponding to the $j$-th output entry of $\psi$ and by $[a]$ the set of integers from 1 to $a$. Based on Lemma 1, we denote by $(\mu_j - \mu_s, \phi_j)$ the ground-truth eigenpairs of the positive semidefinite $\kappa(\boldsymbol{x}, \boldsymbol{x}')$. NeuralEF (Deng et al., 2022) suggests simultaneously solving the following $k$ asymmetric maximization problems will make $\psi_j$ converge to $\phi_j$ for all $j \leq k$:

$$\max_{\psi_j} R_{jj} \quad \text{s.t.:} R_{ij} = 0, c_j = 1, \forall j \in [k], \ i \in [j-1],$$

$$\text{for} \quad R_{ij} \triangleq \iint \psi_i(\boldsymbol{x})\kappa(\boldsymbol{x}, \boldsymbol{x}')\psi_j(\boldsymbol{x}')p(\boldsymbol{x}')p(\boldsymbol{x})d\boldsymbol{x}'d\boldsymbol{x},$$

$$c_j \triangleq \int \psi_j(\boldsymbol{x})\psi_j(\boldsymbol{x})p(\boldsymbol{x})d\boldsymbol{x}.$$

Let $\dot R_{ij} \triangleq \iint \psi_i(\boldsymbol{x})\dot\kappa(\boldsymbol{x}, \boldsymbol{x}')\psi_j(\boldsymbol{x}')p(\boldsymbol{x}')p(\boldsymbol{x})d\boldsymbol{x}'d\boldsymbol{x}$, we have

$$R_{ij} = \dot R_{ij} - \mu_s \iint \psi_i(\boldsymbol{x})\frac{\delta_{\boldsymbol{x}=\boldsymbol{x}'}}{\sqrt{p(\boldsymbol{x})p(\boldsymbol{x}')}}\psi_j(\boldsymbol{x}')p(\boldsymbol{x}')p(\boldsymbol{x})d\boldsymbol{x}'d\boldsymbol{x}$$

$$= \dot R_{ij} - \mu_s \int \left(\int \delta_{\boldsymbol{x}=\boldsymbol{x}'}\psi_j(\boldsymbol{x}')\sqrt{p(\boldsymbol{x}')}d\boldsymbol{x}'\right)\psi_i(\boldsymbol{x})\sqrt{p(\boldsymbol{x})}d\boldsymbol{x}$$

$$= \dot R_{ij} - \mu_s \int \psi_j(\boldsymbol{x})\psi_i(\boldsymbol{x})p(\boldsymbol{x})d\boldsymbol{x}.$$

With the constraint $c_j = \int \psi_j(\boldsymbol{x})\psi_j(\boldsymbol{x})p(\boldsymbol{x})d\boldsymbol{x} = 1$, we have $R_{jj} = \dot{R}_{jj} - \mu_s$, and hence $\max_{\psi_j} R_{jj}$ s.t.: $c_j = 1 \Leftrightarrow \max_{\psi_j} \dot{R}_{jj}$ s.t.: $c_j = 1$. As a result, we can invoke the same proof as in Deng et al. (2022, Appendix A.1) to show that solving $\max_{\psi_1} \dot{R}_{11}$ s.t.: $c_1 = 1$ makes $\psi_1$ converge to $\phi_1$.

Next, we solve the optimization problem for $\dot{R}_{12}$. Under the condition $\psi_1 = \phi_1$, we have

$$
\begin{aligned}
R_{12} =& \dot{R}_{12} - \mu_s \int \psi_1(\boldsymbol{x})\psi_2(\boldsymbol{x})p(\boldsymbol{x})d\boldsymbol{x} \\
=& \iint \psi_1(\boldsymbol{x})\dot{\kappa}(\boldsymbol{x},\boldsymbol{x}')\psi_2(\boldsymbol{x}')p(\boldsymbol{x}')p(\boldsymbol{x})d\boldsymbol{x}'d\boldsymbol{x} - \mu_s \int \psi_1(\boldsymbol{x})\psi_2(\boldsymbol{x})p(\boldsymbol{x})d\boldsymbol{x} \\
=& \int \psi_2(\boldsymbol{x}')p(\boldsymbol{x}') \int \psi_1(\boldsymbol{x})\dot{\kappa}(\boldsymbol{x},\boldsymbol{x}')p(\boldsymbol{x})d\boldsymbol{x}d\boldsymbol{x}' - \mu_s \int \psi_1(\boldsymbol{x})\psi_2(\boldsymbol{x})p(\boldsymbol{x})d\boldsymbol{x} \\
=& \int \psi_2(\boldsymbol{x}')p(\boldsymbol{x}') \int \phi_1(\boldsymbol{x})\dot{\kappa}(\boldsymbol{x},\boldsymbol{x}')p(\boldsymbol{x})d\boldsymbol{x}d\boldsymbol{x}' - \mu_s \int \phi_1(\boldsymbol{x})\psi_2(\boldsymbol{x})p(\boldsymbol{x})d\boldsymbol{x} \\
=& \int \psi_2(\boldsymbol{x}')p(\boldsymbol{x}')\mu_1\phi_1(\boldsymbol{x}')d\boldsymbol{x}' - \mu_s \int \phi_1(\boldsymbol{x})\psi_2(\boldsymbol{x})p(\boldsymbol{x})d\boldsymbol{x} \\
=& \mu_1\langle\phi_1,\psi_2\rangle - \mu_s\langle\phi_1,\psi_2\rangle \\
=& (\mu_1 - \mu_s)\langle\phi_1,\psi_2\rangle,
\end{aligned}
$$

where $\langle\cdot,\cdot\rangle$ denotes the inner product defined as follows:

$$
\langle\varphi,\varphi'\rangle = \int \varphi(\boldsymbol{x})\varphi'(\boldsymbol{x})p(\boldsymbol{x})d\boldsymbol{x} \quad \text{for } \varphi,\varphi' \in L^2(\mathcal{X},p).
$$

Since $\mu_s < 0 < \mu_i, \forall i \in [k-1]$, the constraint $(\mu_1 - \mu_s)\langle\phi_1,\psi_2\rangle = 0$ is equivalent to $\mu_1\langle\phi_1,\psi_2\rangle = 0$. Namely, the constraint $R_{12} = 0$ can be replaced by $\dot{R}_{12} = 0$.

We can apply similar analyses to $R_{ij}, \forall j \in [k], i \in [j-1]$ to show that solving the following $k$ asymmetric maximization problems is equivalent to solving the NeuralEF optimization problems for $\kappa$:

$$
\max_{\psi_j} \dot{R}_{jj} \quad \text{s.t.: } \dot{R}_{ij} = 0, c_j = 1, \forall j \in [k], \ i \in [j-1],
$$

where
$$
\dot{R}_{ij} \triangleq \iint \psi_i(\boldsymbol{x})\dot{\kappa}(\boldsymbol{x},\boldsymbol{x}')\psi_j(\boldsymbol{x}')p(\boldsymbol{x}')p(\boldsymbol{x})d\boldsymbol{x}'d\boldsymbol{x},
$$
$$
c_j \triangleq \int \psi_j(\boldsymbol{x})\psi_j(\boldsymbol{x})p(\boldsymbol{x})d\boldsymbol{x}.
$$

Slacking the constraints on $\dot{R}_{ij}$ as penalties and and implement the constraint on $c_j$ with $L^2$-BN, we obtain Theorem 1.

$\square$

## A.2 PROOF OF EQUATION (7)

*Proof.*

$$R = \mathbb{E}_{p(\boldsymbol{x})}\mathbb{E}_{p(\boldsymbol{x}')}\Big[\kappa(\boldsymbol{x}, \boldsymbol{x}')\psi(\boldsymbol{x})\psi(\boldsymbol{x}')^\top\Big]$$

$$= \mathbb{E}_{p(\boldsymbol{x})}\mathbb{E}_{p(\boldsymbol{x}')}\Big[\frac{p(\boldsymbol{x}, \boldsymbol{x}')}{p(\boldsymbol{x})p(\boldsymbol{x}')}\psi(\boldsymbol{x})\psi(\boldsymbol{x}')^\top\Big]$$

$$= \iint p(\boldsymbol{x}, \boldsymbol{x}')\psi(\boldsymbol{x})\psi(\boldsymbol{x}')^\top d\boldsymbol{x}d\boldsymbol{x}'$$

$$= \mathbb{E}_{p(\boldsymbol{x}, \boldsymbol{x}')}\Big[\psi(\boldsymbol{x})\psi(\boldsymbol{x}')^\top\Big]$$

$$= \mathbb{E}_{p(\bar{\boldsymbol{x}})}\mathbb{E}_{p(\boldsymbol{x}|\bar{\boldsymbol{x}})p(\boldsymbol{x}'|\bar{\boldsymbol{x}})}\Big[\psi(\boldsymbol{x})\psi(\boldsymbol{x}')^\top\Big]$$

$$\approx \frac{1}{b}\sum_{i=1}^{b}\mathbb{E}_{p(\boldsymbol{x}|\bar{\boldsymbol{x}}_i)p(\boldsymbol{x}'|\bar{\boldsymbol{x}}_i)}\Big[\psi(\boldsymbol{x})\psi(\boldsymbol{x}')^\top\Big]$$

$$\approx \frac{1}{b}\sum_{i=1}^{b}\psi(\boldsymbol{x}_i)\psi(\boldsymbol{x}_i^+)^\top,$$

where $\bar{\boldsymbol{x}}_i$ are samples from $p(\bar{\boldsymbol{x}})$ and $\boldsymbol{x}_i$ and $\boldsymbol{x}_i^+$ are two independent samples from $p(\boldsymbol{x}|\bar{\boldsymbol{x}}_i)$. $\qquad\square$

## B MORE RESULTS

### B.1 DISCUSSION ON THE ARCHITECTURE OF THE PROJECTOR FOR VISUAL REPRESENTATION LEARNING

The projector trick is widely used in SSL (He et al., 2020; Chen et al., 2020a). We follow the trend and add an MLP projector after the ResNet-50 encoder for representation learning on ImageNet. We empirically diagnose the MLP projector and find that, when removing the MLP projector or replacing it with a linear one or removing the BN after the hidden layer, Neural Eigenmap failed to converge or performed poorly. This finding is partially consistent with the results in some SSL works (e.g., Chen & He, 2021) and we conclude that an MLP projector with BNs in the hidden layer plays an important role in the success of Neural Eigenmap for visual representation learning.

### B.2 ORTHOGONALITY OF THE LEARNED EIGENFUNCTION APPROXIMATIONS

While it is challenging to directly verify the orthogonality and accuracy of the learned eigenfunctions for the augmentation kernel, primarily due to the unavailability of data distribution densities used to define this kernel, we conducted an additional experiment on the RBF kernel, which is more amenable to analysis. We include the results in Figure 4, where we plot the learned eigenfunction approximations alongside the ground truth. We also conducted an orthogonality check. As shown, our learned approximations are accurate.

We consider Figures 2 and 6 as indirect evidence supporting that the learned eigenfunction approximations of the augmentation kernel are accurate (such that the features are ordered by eigenvalues).

### B.3 COMPARE NEURAL EIGENMAP TO SCL+PCA FOR IMAGE RETRIEVAL

For the considered image retrieval task, it is a straightforward idea to apply PCA to SCL's representations to induce structures. Then, we can select features according to the index to get adaptive-length codes for image retrieval, as done in Neural Eigenmap. In this subsection, we test this proposal. Specifically, based on SCL, we compute the principal components of ImageNet training set feature covariances and use the components to project image features into a $k$-dimensional eigenspace for retrieval. Table 7 and Table 8 display the performance comparison between this method and our Neural Eigenmap.

As shown, our method outperforms this PCA-based approach for short code lengths, suggesting that combining SCL with PCA is not optimal for recovering principal eigenfunctions of the augmentation

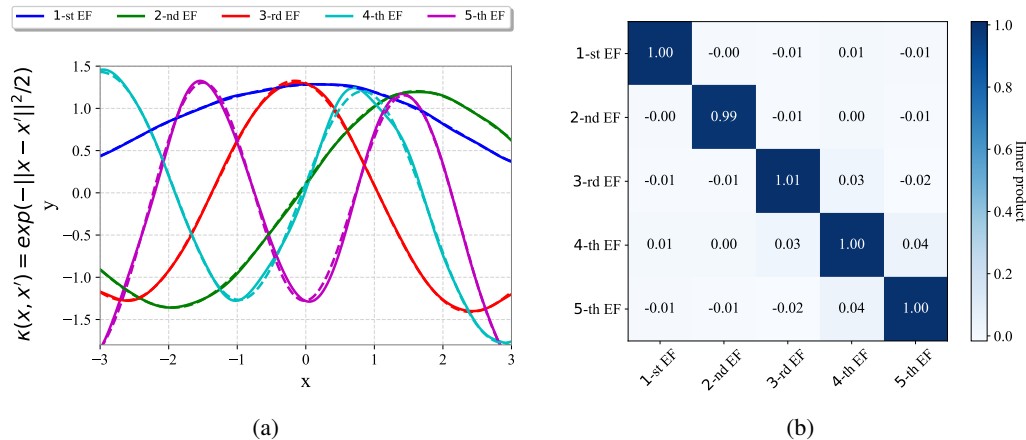

(a)                                            (b)

Figure 4: Visualization of the learned neural eigenfunctions (EFs) by our approach for the RBF kernel. The dashed lines represent the ground-truth eigenfunctions. We also provide the inner products between them to verify the orthogonality. We use a 3-layer MLP of width 256 to define the neural eigenfunctions in this experiment. We train on randomly sampled 1024 positions in the range but test on uniformly sampled 2048 positions.

Table 7: Comparison of retrieval mAP on NUS-WIDE.

| Code length | 4 | 8 | 16 | 32 |
|---|---|---|---|---|
| Our | 0.6706 | 0.7213 | 0.7401 | 0.7446 |
| SCL+PCA | 0.4845 | 0.6679 | 0.7368 | 0.7579 |
| SCL | 0.5439 | 0.5866 | 0.5909 | 0.6207 |

Table 8: Comparison of retrieval mAP on COCO.

| Code length | 4 | 8 | 16 | 32 |
|---|---|---|---|---|
| Our | 0.5934 | 0.6420 | 0.6657 | 0.6832 |
| SCL+PCA | 0.4819 | 0.5645 | 0.6360 | 0.6902 |
| SCL | 0.4995 | 0.5714 | 0.5572 | 0.5882 |

kernel. Besides, we would like to point out that applying PCA as a post-processing step to self-supervised learning methods would substantially increase the computational burden (e.g., it needs to store the principal components apart from the network), which contradicts our paper's goal of avoiding expensive nonparametric approaches.

### B.4 MORE VISUALIZATION OF THE LEARNING EIGENFUNCTIONS

To further explore the visualization of the learned eigenfunctions on ImageNet for the augmentation kernel, we optimize the input starting from random noise to maximize the function output. To enhance interpretability, we incorporate Gaussian blur in the input, facilitating the emergence of patterns recognizable by humans. The optimization results for the first 8 principal neural eigenfunctions are shown in Figure 5. Although precise information may be challenging to discern from these visualizations, we discover that different eigenfunctions exhibit distinct pattern preferences. This finding aligns with our original intentions.

### B.5 OTHER VISUALIZATIONS

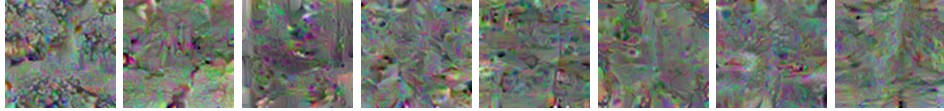

Figure 5: The optimized images that maximize the output of the first 8 principal neural eigenfunctions. The optimization starts from random noise, and the inputs are augmented by Gaussian blur to encourage the emergence of human-identifiable patterns.

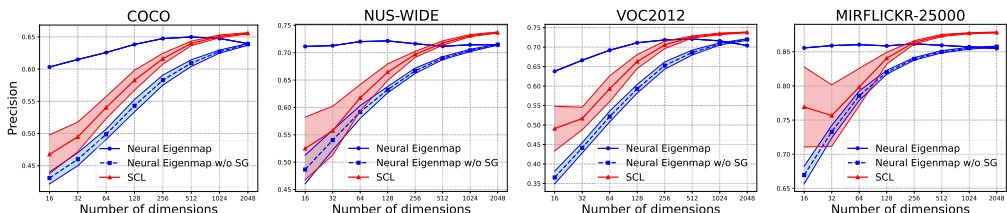

Figure 6: Retrieval precision varies w.r.t. representation dimensionality.

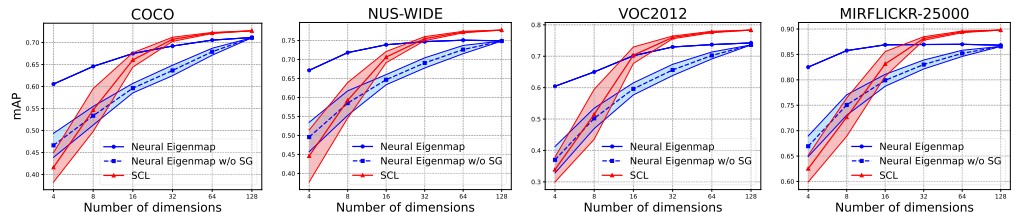

Figure 7: Retrieval mAP varies w.r.t. representation dimensionality when using a small $k$ (the hidden dimension of the projector is 8192 while the output dimension is 128).

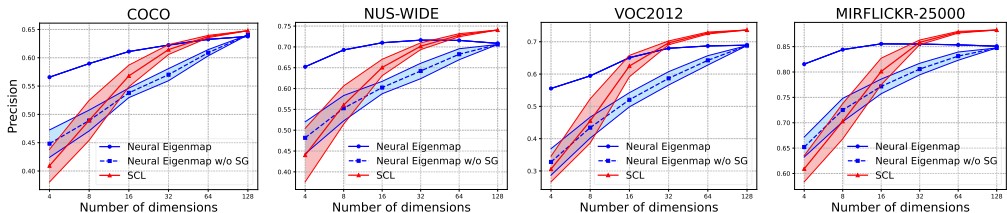

Figure 8: Retrieval precision varies w.r.t. representation dimensionality when using a small $k$ (the hidden dimension of the projector is 8192 while the output dimension is 128).

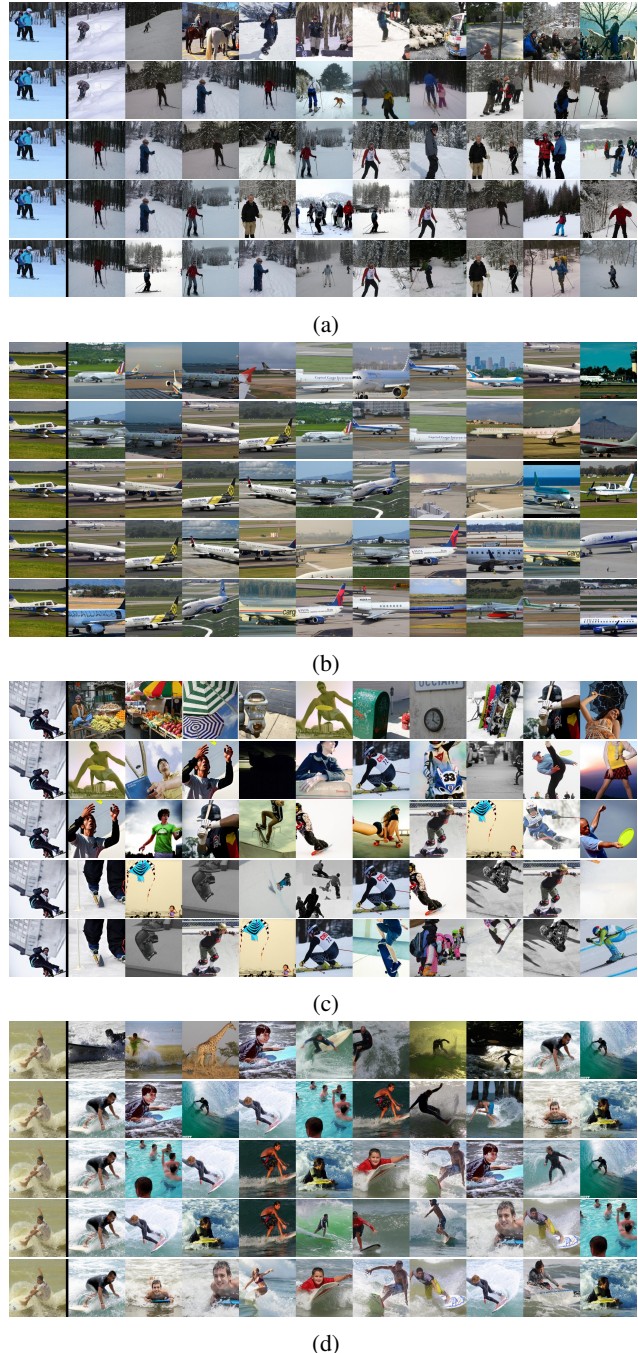

(a)

(b)

(c)

(d)

Figure 9: Visualization of retrieval results on COCO with the representations yielded by Neural Eigenmap. The five rows correspond to using the first 4, 8, 16, 32, and 64 entries of the projector outputs for retrieval, respectively. In each row, the first image is a query, and the rest are the top 10 images closest to it over the set.

