# OpenReview forum: "Neural Eigenfunctions Are Structured Representation Learners"
_ICLR.cc/2024/Conference — Submitted to ICLR 2024_

### Official Review · Reviewer_RwzV · 2023-10-29

**Soundness:** 3 good
**Presentation:** 4 excellent
**Contribution:** 3 good
**Rating:** 6
**Confidence:** 4

**Summary:**

Based on NeuralEF, author(s) proposed the Neural Eigenmap that is a structured and adaptive deep presentation. Under the new formulation, an objective function can resemble those popular self-supervised learning methods. The benefits that the Neural Eigenmap can offer is that the learned representations are structures which preserve the most important information from the data. At the same time the Neural Eigenmap has been extended for graph data, building on a theoretical result.

**Strengths:**

1. Clearly describe the relations among Neural Eigenmap, SSL methods, and NeuralEF.
2. Although it builds on the combination of NeuralEF and SSL Kernel, the benefits from there are clearly derived.
3. The other contribution is to generalize the Neural Eigenmap for graph data. The new formulation can be scaled up for larger graph datasets.

**Weaknesses:**

The paper is well presented with clear contributions in a very simple form.

**Questions:**

Thanks for providing the accompany code for the algorithm.

1. I have tracked the training process over graph dataset arXiv. It seems to me that there is no special ways to take the augmented nodes. It seems to me it quite arbitrarily to construct X^+, although adjacency matrix is applied.

2. Not sure why the first term in (12) is squared in the code.

---

> ### Author Response · Authors · 2023-11-19
> **Thanks for your constructive comments**
>
> We appreciate the positive comments, and we are encouraged by the recognition of our presentation and contribution. We address the detailed comments below.
>
> ### Regarding code
>
> We clarify that our graph representation learning objective Eq. (12) is different from the SSL objective in Section 3. Specifically, instead of utilizing the argumentation kernel, we employed the graph adjacency kernel. As a result, data augmentation (represented as $X^+$) is not needed.
>
> We also clarify that we did not square the first term in Eq. 12. The code for computing this term is in L75 of code/graph/main_products.py.

---

> > ### Comment · Reviewer_RwzV · 2023-11-22
> >
> > Thanks for the clarification

---

> > > ### Author Response · Authors · 2023-11-22
> > > **Thanks**
> > >
> > > Thanks for your time! We will continue to update the manuscript.
> > >
> > > Best,
> > > The Authors

---

### Official Review · Reviewer_t4xz · 2023-10-31

**Soundness:** 3 good
**Presentation:** 3 good
**Contribution:** 3 good
**Rating:** 6
**Confidence:** 3

**Summary:**

The paper proposes the use of NeuralEF, a neural network that aims to learn the eigenfunctions of a linear operator from large-scale data, for purposes of parametric representation learning; this is termed as a neural eigenmap. The paper provides connections between the formulation of the proposed approach and those of existing self-supervised learning methods. The paper also shows the application of the method to settings with indefinite kernels for the affinity matrix.

**Strengths:**

The paper leverages a recently proposed approach for eigenfunction estimation in a commonly used application of eigendecompositions for learning. The reasoning is straightforward and the results are compelling.

**Weaknesses:**

Some of the choices made are not clearly explained. For example, the choice of enabling/disabling stop_grad is said to allow/not allow for ordered eigenfunctions/structured representations, but there is no discussion of this (e.g., what does stop_grad do). This is also implied in the choice of elements with small indices vs. random elements being chosen when evaluating these two approaches.

The proposed method is similar in Formulation to Barlow Twins, but the numerical comparison is focused on a few experiments.

Minor comments:

* Just before eq. 3, define BN layers (as batch normalization layers?)
* Section 4: "principle eigenfunctions" should be "principal eigenfunctions".

**Questions:**

Can the optimization in (3) be connected to the original definition of the eigenfunction from (1)?

Can the authors elaborate on how the ordered structure arises in NeuralEF? Particularly given that the orthogonality requirement for eigenvectors in an eigendecomposition is not present in the NeuralEF formulation (8) to obtain the set of "learned" eigenvectors.

After eq. 10, should $\gamma$ be $\lambda$?

Is there intuition as to why (9) works better than (10)?

Can the authors elaborate on the scalability of the proposed approach vs. that of Johnson et al. (2022)?

Section 6.2 first sentence refers to Section 6.2 - is that a typo?

---

> ### Author Response · Authors · 2023-11-19
> **Thanks for your constructive comments**
>
> Thank you for taking the time to review our paper. We are glad that you found our reasoning straightforward and our results compelling. We have provided responses to the detailed questions below.
>
> ### Regarding enabling/disabling stop_grad
>
> We first apologize for typos in Eq 3 of the original manuscript and have revised it accordingly. We clarify that the second term $R_{i,j}^2$ in Eq 3 is asymmetric—inducing updates for only $\\psi_j$ instead of $\\psi_i, i< j$. Therefore, when optimizing the $k$ problems simultaneously using a shared neural network $\\psi$,  we obtain the “surrogate” loss in Eq 8, where the stop_grad operation implements such asymmetry. In the presence of the stop_grad, the learning strictly follows Eq 3, and the learning outcomes are ordered eigenfunctions, so we obtain structured representations where choosing elements with small indices corresponds to choosing eigenfunctions with large eigenvalues (the “principal” components).
>
> We remove the stop_grad operation for linear probe evaluation. This is because the linear probe does not take advantage of ordered representations (as shown in HaoChen et al., 2021, Lemma 3.1), while such a replacement may actually ease optimization because it relaxes the ordering constraints. Yet, after doing so, the ordered structure of the learned representation vanishes, so to perform image retrieval, we have to randomly select elements for an equal-length comparison and perform multiple runs.
>
> ### Similarity in formulation to Barlow Twins while numerical comparison focused on a few experiments.
>
> As explained in the paragraph **Connection to Barlow Twins**, the objectives of the two approaches are similar but the gradients and optimal solutions are different. The Neural Eigenmap objective is theoretically grounded and is guaranteed to recover the eigenfunctions span, while Barlow Twins can be seen as a less-principled approximation to this objective. Additionally, Barlow Twins lacks the ability to generate ordered representations and handle graph representation learning. For the empirical evaluation, we have shown statistically significant improvements over Barlow Twins across three tasks (ImageNet linear probe, COCO detection, and instance segmentation), a variety of batch sizes and projector dimensions. We believe these gains are meaningful and kindly ask the reviewer to reconsider their evaluation of these results.
>
> ### Define BN layers and other typos
>
> Thanks. We have revised the paper accordingly.
>
> ### Can the optimization in Eq 3 be connected to the original definition of the eigenfunction from Eq 1?
>
> Yes, as proved in the original NeuralEF paper (Deng et al. 2022, Theorem 1), the optima of the optimization problems in Eq 3 are the principal eigenfunctions, satisfying Eq 1.
>
> ### How does the ordered structure arise in NeuralEF? Orthogonality requirement not present in (8)
>
> We clarify that although the orthogonality requirement is not explicitly enforced in Eq. 8 (which is derived from the NeuralEF problem in Eq. 3), the orthogonality is automatically achieved at the optimum of the problem.
>
> Intuitively speaking, the first term in Eq 3 corresponds to finding the principal eigenvectors by maximizing the Rayleigh quotient. When estimating multiple eigenvectors simultaneously, we also need to ensure that the $j$-th eigenvector is orthogonal to the $i$-th eigenvector for all $i < j$. The second term in Eq 3 is a penalty added to achieve this, as proven in the original NeuralEF paper (Deng et al., 2022).
>
> ### $\\gamma$ should be $\\lambda$
>
> Thanks. Yes, it is a typo. We have updated the manuscript.
>
> ### Why Eq 9 works better than Eq 10
>
> We believe Equation 9 works better than Equation 10 because of its theoretical underpinnings, ensuring the recovery of the eigenfunction span at the optimum. Conversely, we view Equation 10 as an “incorrect way” to implement the same idea.
>
> ### The scalability of the proposed approach vs. that of Johnson et al. (2022)?
>
> Our approach is as scalable as regular self-supervised learning methods—we demonstrate this on ImageNet-scale experiments and a graph representation learning benchmark with more than one million nodes, while Johnson et al. (2022) rely on the nonparametric kernel PCA, which has a cubic complexity w.r.t. the number of training points. Their experiments were conducted on 2D-synthetic datasets and MNIST.
>
> ### Section 6.2 first sentence refers to Section 6.2
>
> We apologize for the typo. We have revised the manuscript.

---

### Official Review · Reviewer_WyYb · 2023-11-02

**Soundness:** 3 good
**Presentation:** 3 good
**Contribution:** 3 good
**Rating:** 8
**Confidence:** 3

**Summary:**

In this work, the authors propose a principled theoretical motivation for eigenfunction based representation learning in the form of Neural Eigenmap and show it as providing a unifying surrogate for unsupervised representation learning. They show that this objective is close to the self-supervised learning objectives with a symmetry breaking property and present an approach to optimize it in a parametric manner, thus enabling scaling unlike many of the previous approaches-- showing that it is possible to learn eigenfunctions for a large dataset like ImageNet, along with OOD generalisation.

**Strengths:**

- Through this work, the authors provide additional arguments for learning the principle eigenfunctions for unsupervised representation learning by considering the integral operator of a pre-defined kernel and the data distribution, which other lines of work in this area have argued to be at the core of many machine learning problems.
- Over pre-existing work such as Johnson et al and Haochen et al, Neural Eigenmap shows better scalability with the number of eigenfunctions, and learning of an ordered structure in the representation. Also, specific to graph representation learning, the proposed method has faster forward and backward passes.
- Neural Eigenmap shows similar retrieval performance such as Barlow Twins (BT) and Spectral Contrastive Learning (SCL) on image retrieval benchmarks such as COCO etc with much fewer representation dimensions-- the authors show this is 16x fewer.

**Weaknesses:**

- Maybe not a weakness, but in this kind of learning, how do we know which kernel to pre-define?

* With respect to the linear probe experiments for unsupervised representation learning, a few questions-
    * Why have the authors reproduced Barlow Twins themselves when, if I am not mistaken, the top-1 accuracy for ImageNet with a ResNet-50 pretrained encoder is available (like it is for SCL, which the authors use)?
    * Can you make any inferential comments on the link between batch size, number of epochs, projector dimensions?
    * For Barlow Twins, the authors in that paper report that the model has highest top-1 accuracy at a batch size of 1024, so like why are the results selected for a batch size of 2048 in this paper?

**Questions:**

- Maybe something I am missing: while talking about the connection between PCA and the proposed method in the subsection on “Learning ordered representations”, I have a general doubt— isn’t PCA a special, linear case of spectral analysis on Euclidean space— and hence follows the observation on ordering and the principled components carrying the most information, or is there a deeper connection to talk about here?

- Something I didn’t understand: why does replacing $\hat{\psi_{X_B}}$ by $\psi_{X_B}$ not affect the optimal classifier? (when talking about the linear probe evaluation)

* What are the benchmarks that ignore, and that which do not ignore feature importance? Are they based on the task of image retrieval?

* The authors remark that the proposed method has faster forward/backward passes than graph neural networks which have a computational complexity of $O(n^3)$-- what is the computational complexity of this method?

**Details Of Ethics Concerns:**

I don't think there are any ethics concerns for this paper.

---

> ### Author Response · Authors · 2023-11-19
> **Thanks for your constructive comments**
>
> We appreciate the positive comments, and we are encouraged by the recognition of our contributions compared to existing works and the empirical effectiveness of the proposal. We address the detailed concerns below.
>
> ### Which kernel to predefine
>
> The purpose of our Section 3 and 4 is to address the question. In Section 3, along with the accompanying experiments, we showed strong evidence that the augmentation kernel is an excellent option for visual representation learning. Furthermore, in Section 4 and 6.3, we demonstrated good performance can be achieved by graph adjacency kernels in graph representation learning tasks.
>
> ### Regarding the linear probe experiments for unsupervised representation learning
>
> *Why have the authors reproduced Barlow Twins themselves?* This is because the linear probe results in the paper of Barlow Twins correspond to 1000-epoch training, while we perform 100-epoch training (as done in the paper of SCL) for results in Table 1 due to limited resources.
>
> *Can you make any inferential comments on the link between batch size, number of epochs, and projector dimension?* Generally, there are no trade-offs between these three factors. For our method, batch size and projector dimension need both to be large enough for good linear probe performance, regardless of the number of epochs. A larger batch size contributes to a more reliable Monte Carlo estimation of the expectation in Eq 7, and a higher projector dimension translates to more eigenfunctions involved in the representation. For other self-supervised methods, different batch sizes and projector dimensions are preferred by different methods, e.g., contrastive learning methods without memory buffers may need a large batch size while others may not; SimCLR is robust to projector dimension, while Barlow Twins needs large projector dimensions.
>
> *Why are the results of Barlow Twins of a batch size of 2048 reported?* We perform a new set of experiments for Barlow Twins using a batch size of 1024 and projector dimension of 8192, and obtain 65.5±0.1% top-1 linear probe accuracy on ImageNet, slightly weaker than using a batch size of 2048 for the projector dimension of 8192. The gap can stem from that we have not carefully tuned the learning rate for the LARS optimizer, but it is not significant, which echoes the results in Barlow Twins’ paper that it is relatively robust against batch sizes from 512 to 2048.
>
> ### Connection between PCA and the proposed method
>
> Your understanding is totally correct. We use such an analogy to intuitively explain why our learned representation is structured and ordered.
>
> ### Why replacing $\\hat{\\psi}$ by $\\psi$ does not affect the optimal classifier for linear probe
>
> Such a replacement makes the learned representation converge to the output of an arbitrary span of eigenfunctions. It amounts to applying a right transformation on the original eigenmaps. As proven by Lemma 3.1 of HaoChen et al., (2021), the effect of such a transformation can be easily eliminated by a trivial manipulation of the weight matrix of the linear classifier.
>
> ### When does feature importance matter?
>
> All benchmarks that use linear probe evaluations (Sections 6.2 and 6.3) ignore the significance of individual features. However, the image retrieval experiment in Section 6.1 utilizes the importance of features as they affect the tradeoff between cost and quality. By truncating less important features, our method can achieve similar retrieval quality using a much shorter code length than other methods that lack feature ordering.
>
> ### Computational complexity for graph data
>
> We first clarify that we did not state that graph NNs have a computational complexity of $O(n^3)$. Instead, it refers to classic nonparametric graph embedding methods like Laplacian Eigenmaps that involve matrix decomposition, which do have that complexity. Due to stochastic training, the computational complexity for our training is linear w.r.t. the number of data points $n$. During inference, when computing the embedding of a single node, our method simply requires forward propagation of a neural network with the features of that specific node as input, while graph NNs require aggregating information from the neighborhood on graphs. This neighborhood can be large, and hence the cost may be high.

---

### Official Review · Reviewer_oHVo · 2023-11-03

**Soundness:** 2 fair
**Presentation:** 2 fair
**Contribution:** 2 fair
**Rating:** 6
**Confidence:** 3

**Summary:**

The authors propose a method to learn the eigenfunctions of an augmentation kernel or a given graph, by parametrizing them using deep networks, which can be used for unsupervised representation learning. Compared to previous related works the proposed approach is scalable, while it can provide ordered eigenfunctions. In the experiments, it is demonstrated the performance of the approach on downstream tasks.

**Strengths:**

- Unsupervised representation learning is an interesting problem, while the proposed approach extends previous works.
- The technical part of the paper seems to be correct, but I have not checked in detail all the theoretical results.
- The proposed methods perform better in some of the experiments.

**Weaknesses:**

- The paper is ok written, but there are some parts that need improvement. For example, it is not entirely clear which methods the authors propose and how they differ from previous works.
- Regarding the novelty, it is unclear to me what is the difference between the proposed approaches compared to previous works, which makes hard to understand the actual contributions.
- In some of the experiments, it seems that the improvement is not significant.

**Questions:**

Q1. Regarding the graph-based problem (Sec. 4) it is clear that Eq. 12 is the proposed model, and due to $\hat{\psi}$ in the second term, the learned eigenfunctions are ordered. I think in Sec. 3 this is not entirely clear. As far as I understand, Eq. 8 which gives ordered representations is already proposed by Jonson et al. (2022)? While the proposed Eq. 9 does not give ordered representations but it is potentially scalable?

Q2. I think that it is not clear from the text when the $\hat{\psi}$ in the second term of Eq. 8 should be used and when not.

Q3. I believe that the analysis of the comparison to Barlow Twins is rather limited. It seems that the approaches are very similar both in their formulation but also in the experiments. I believe the differences should be investigated in more detail.

Q4. As far as I understand, both SCL and Neural Eigenmaps learn the top-k eigenfunctions of a kernel. Why there is a difference in performance e.g. Fig 2? Neural eigenmaps approximates better the true eigenfunctions compared to SCL?

Q5. There is an example in the appendix that shows the approximation of the eigenfunctions for the RBF kernel. Similar to the spectral methods, I think that is interesting to see a comparison using 3D shapes, e.g. the Standford Bunny.

---

> ### Author Response · Authors · 2023-11-19
> **Thanks for your constructive comments**
>
> We appreciate the constructive feedback. Below, we address the detailed concerns.
>
> ### Clarification regarding Johnson et al. (2022)
>
> In reference to the extended conference version [1] of Johnson et al.'s (2022) paper, we see that they also investigated using NeuralEF as an alternative to their main kernel PCA approach. However, we want to emphasize that this should be considered as **concurrent to our work**, as evident from our submission history. We will make this more clear in the paper. Additionally, it's important to note that the two papers have distinct focuses. [1] primarily investigates the optimality of representations obtained through kernel PCA. It only mentioned NeuralEF as an alternative and tested it in synthetic tasks. In contrast, our work extends the application of NeuralEF to larger-scale problems such as ImageNet-scale SSL and graph representation learning. Notably, [1] did not discuss the benefit of ordered representation while we clearly demonstrated it in the image retrieval tasks.
>
> [1] Johnson, D. D., El Hanchi, A., & Maddison, C. J. Contrastive Learning Can Find An Optimal Basis For Approximately View-Invariant Functions. In International Conference on Learning Representations, 2023.
>
> ### When the $\\hat{\\psi}$ in the second term of Eq. 8 should be used and when not
>
> We recommend the use of $\\hat{\\psi}$ when downstream tasks require an ordered representation. However, in cases where the order of features has no impact, it is preferable to use $\\psi$. In our paper, we employed two distinct tasks to demonstrate this concept. In Section 6.2, when evaluating visual representation learning through linear probe, we used $\\psi$ in the second term. This choice was due to the fact that the optimal classifier for this task is unaffected by the ordering of eigenfunctions (as shown in HaoChen et al., 2021, Lemma 3.1). Conversely, for the image retrieval task discussed in Section 6.1, we employed $\\hat{\\psi}$ in Eq. 8. This allowed us to enforce feature ordering, enabling us to achieve the best quality-cost tradeoff (Figure 2) by discarding less important features.
>
> ### Similarity to Barlow Twins
>
> As explained in the paragraph **Connection to Barlow Twins**, the objectives of the two approaches are similar but the gradients and optimal solutions are different. The Neural Eigenmap objective is theoretically grounded and is guaranteed to recover the eigenfunctions span, while **Barlow Twins can be seen as a less-principled approximation** to this objective. Apart from that, Barlow Twins cannot yield ordered representations and handle graph representation learning. For the empirical evaluation, we have included comparisons to Barlow Twins in Tables 1-4. The results show statistically significant improvements over Barlow Twins across several tasks, batch sizes, projector dimensions, and datasets. We believe these gains are meaningful and kindly ask the reviewer to reconsider their evaluation of these results.
>
> ### Difference between SCL and Neural Eigenmap
>
> We clarify that **SCL does not directly compute the top-k eigenfunctions of a kernel**. Rather, SCL recovers the subspace spanned by the eigenfunctions, without determining their specific order or ranking. As explained in HaoChen et al. (2021) in the paragraph before Lemma 3.1, SCL learns the eigenfunctions up to a right transformation. Consequently, the representation learned by SCL does not exhibit an ordered structure as ours, which results in the performance gap in Fig 2.
>
> ###  Using 3D shapes, e.g. the Standford Bunny, for showing the approximation of the eigenfunctions
>
> Thanks for the advice. Although the primary focus of this paper is not the approximation accuracy of NeuralEF, we will try offering visualizations of the learned eigenfunctions for 3d shapes in the final version. In the meantime, we kindly direct the reviewer to the original NeuralEF paper (Deng et al, 2022) for approximation results on 3d shapes like swiss rolls.

---

> > ### Author Response · Authors · 2023-11-21
> > **Looking forward to further discussions**
> >
> > Dear reviewer,
> >
> > We are wondering if our response and revision have resolved your concerns. If so, we would highly appreciate it if you could re-evaluate our work and consider raising the score.
> >
> > If you have any additional questions or suggestions, we would be happy to have further discussions.
> >
> > Best regards,
> >
> > The Authors

---

> > > ### Author Response · Authors · 2023-11-22
> > > **Are there any further questions?**
> > >
> > > Dear Reviewer,
> > >
> > > Thank you for dedicating your time to reviewing our paper. As the deadline for the rebuttal is approaching, we would like to inquire if there are any unresolved questions or concerns. If there are none, we kindly request that you consider raising the score.
> > >
> > > Best regards,
> > > The Authors

---

> > > > ### Comment · Reviewer_oHVo · 2023-11-22
> > > > **Post-rebuttal**
> > > >
> > > > I would like to thank the authors for the answers. After reading the rest of the reviews and the associated answers, I think that the paper seems to be within the standards of the conference, so I will keep my score 6. However, I recommend the authors to update the paper by taking into account the comments of the reviewers and include further clarifications in the relevant parts.

---

> > > > > ### Author Response · Authors · 2023-11-22
> > > > > **Reply**
> > > > >
> > > > > Thank you for your response and kind suggestion. We will certainly incorporate the rebuttal into the revision. By the way, it appears that your score is 5... Would you mind providing an update on this?
> > > > >
> > > > > Best regards,
> > > > > The Authors

---

### Meta-Review · Area_Chair_MM3M · 2023-12-06

**Metareview:**

The paper proposes an approach, Neural Eigenmap, for parametric representation learning using a recent method NeuralEF, a neural network for approximating principal eigenfunctions of kernels, definite as well as indefinite for graph representation learning problems.

The reviewers appreciated the added structure offered by the proposed method, yielding in particular leading eigenfunctions. However, based on the reviews as well as internal discussions post-rebuttal, the method was largely perceived as a relatively direct application of NeuralEF, in part due to certain resemblance of the resulting formulation to existing dominant ones (in SSL) as well as the somewhat incremental improvement in the performances demonstrated on large-scale benchmarks. Despite the discussions with the authors, evaluations generally remained borderline. The authors are encouraged to incorporate the important feedback given by the knowledgeable reviewers.

**Justification For Why Not Higher Score:**

Incremental rather direct use of existing components, but can be bumped up.

**Justification For Why Not Lower Score:**

N/A

---

### Decision · Program_Chairs · 2024-01-16

Reject